# Association between Early Phase Serum Lactate Levels and Occurrence of Delayed Neuropsychiatric Sequelae in Adult Patients with Acute Carbon Monoxide Poisoning: A Systematic Review and Meta-Analysis

**DOI:** 10.3390/jpm12040651

**Published:** 2022-04-18

**Authors:** Heekyung Lee, Jaehoon Oh, Hyunggoo Kang, Chiwon Ahn, Myeong Namgung, Chan Woong Kim, Wonhee Kim, Young Seo Kim, Hyungoo Shin, Tae Ho Lim

**Affiliations:** 1Department of Emergency Medicine, College of Medicine, Hanyang University, Seoul 04763, Korea; massdt@hanyang.ac.kr (H.L.); ojjai@hanyang.ac.kr (J.O.); newnowold@hanyang.ac.kr (H.S.); erthim@hanyang.ac.kr (T.H.L.); 2Department of Emergency Medicine, College of Medicine, Chung-Ang University, Seoul 06974, Korea; cahn@cau.ac.kr (C.A.); myeong15180@caumc.or.kr (M.N.); whenever@cau.ac.kr (C.W.K.); 3Department of Emergency Medicine, Hallym University, Chuncheon 24252, Korea; wonsee02@gmail.com; 4Department of Neurology, College of Medicine, Hanyang University, Seoul 04763, Korea; kimys1@hanyang.ac.kr

**Keywords:** lactate, carbon monoxide poisoning, delayed neuropsychiatric sequelae, meta-analysis, biomarker

## Abstract

The primary goal of treating carbon monoxide (CO) poisoning is preventing or minimizing the development of delayed neuropsychiatric sequelae (DNS). Therefore, screening patients with a high probability for the occurrence of DNS at the earliest is essential. However, prognostic tools for predicting DNS are insufficient, and the usefulness of the lactate level as a predictor is unclear. This systematic review and meta-analysis investigated the association between early phase serum lactate levels and the occurrence of DNS in adult patients with acute CO poisoning. Observational studies that included adult patients with CO poisoning and reported initial lactate concentrations were retrieved from the Embase, MEDLINE, Google Scholar and six domestic databases (KoreaMED, KMBASE, KISS, NDSL, KISTi and RISS) in January 2022. Lactate values were collected as continuous variables and analyzed using standardized mean differences (SMD) using a random-effect model. The risk of bias was evaluated using the Quality in Prognosis Studies (QUIPS) tool, and subgroup, sensitivity and meta regression analyses were performed. Eight studies involving a total of 1350 patients were included. The early phase serum lactate concentration was significantly higher in the DNS group than in the non-DNS group in adult patients with acute CO poisoning (8 studies; SMD, 0.31; 95% CI, 0.11–0.50; I^2^ = 44%; *p* = 0.002). The heterogeneity decreased to I^2^ = 8% in sensitivity analysis (omitting Han2021; 7 studies; SMD, 0.38; 95% CI, 0.23–0.53; I^2^ = 8%; *p* < 0.001). The risk of bias was assessed as high in five studies. The DNS group was associated with significantly higher lactate concentration than that in the non-DNS group.

## 1. Introduction

Carbon monoxide (CO) remains one of the most common causes behind the number of poisoning admissions to the emergency departments (EDs), with an incidence of 137 cases and 4.6 deaths per million individuals worldwide [1]. More than 50,000 cases are reported annually in the United States, and 24,890 deaths have occurred between 1999 to 2014 [2,3]. According to the World Health Organization, more than 140,000 people died due to CO poisoning between 1980 and 2008 in 28 European member countries [4]. CO is produced upon burning carbon-containing fuel in an oxygen-deficient state. The incomplete combustion of carbon compounds by indoor heaters, fire and smoke in the case of a fire incidence, and automobile exhaust are common sources of CO [5,6]. Although the symptoms of CO poisoning are non-specific, exposure to low levels may cause headaches, dizziness, and neuropsychiatric abnormalities. Moreover, moderate to severe exposure can lead to confusion, loss of consciousness and even death [7].

The binding affinity of CO molecules to hemoglobin is 200 times stronger than that of oxygen after inhalation, producing a stable complex called carboxyhemoglobin (COHb). The generated COHb causes cellular hypoxia by shifting the oxyhemoglobin dissociation curve to the left, reducing the oxygen release, and increasing the cytosolic heme concentration, causing oxidative stress [7,8]. CO binds to the heme protein, disrupts cellular respiration, and causes neuronal necrosis and apoptosis by producing reactive oxygen species [9,10,11,12]. These mechanisms lead to systemic toxic effects caused by tissue ischemia, local inflammatory response and nerve cell damage, where damage to the cardiovascular system and neurological damage appear to be major pathologies of concern [7,8].

In addition to acute symptoms and complications, CO poisoning leads to a subacute and chronic complication called delayed neuropsychiatric sequelae (DNS) [13,14]. Although the mechanism of DNS development is unclear, diffuse inflammation is observed in the deep white matter and periventricular area [15,16]. The main symptoms of DNS vary from psychotic symptoms, such as depression, insomnia and anxiety, to neuropathic symptoms, such as headache, dizziness, gait disturbance and cognitive and disorientation disorders [14,17]. After recovery from acute poisoning, the development of DNS can occur for up to 1 year. Most patients develop DNS within 6 weeks, and DNS is seen in 3 to 40% of patients with CO poisoning [17,18,19,20,21,22,23,24]. Although the preventive effect of hyperbaric oxygen therapy (HBOT) on DNS is debatable, it could be helpful to use HBOT in patients with CO poisoning as soon as possible to prevent DNS and reduce its severity [25].

Several studies have investigated tools for predicting the occurrence of DNS in patients with acute CO poisoning [17,18,19,20,21,22,23,24]. However, standard screening tools or standardized guidelines to accurately predict DNS development are lacking. COHb is used as a standard indicator to diagnose CO poisoning and to confirm the severity of CO poisoning; however, it does not help predict DNS [17,26,27]. The elimination half-life of COHb in CO poisoning patients treated with 100% oxygen at normobaric pressure is about 1 h [28,29]. COHb levels fall rapidly after the end of CO exposure and are decreased before being determined at the ED. Since the major effect of CO is systemic hypoxia, accompanied by an increase in lactate concentration, elevated serum lactate concentration can be used as an indicator of systemic hypoxia [30,31,32]. Accordingly, several previous studies have been conducted to compare the initial lactate concentration in DNS and non-DNS development groups, where inconsistent results have been reported, with remarkable differences [17,18,19,20,21,22,23,24]. Pepe et al. reported that there was no significant difference in early phase lactate levels between DNS and non-DNS groups (mmol/L; 1.77 vs. 1.76, *p* > 0.05) [17]. On the contrary, in a retrospective observational study, the crude odds ratio of the initial lactate level for development of DNS was reported to be 1.10 (*p* = 0.005) [20].

The aim of treating acute CO poisoning is to prevent and reduce the occurrence of DNS and the early identification of patients at a high risk of DNS development. However, tools for accurately predicting DNS are insufficient, and the predictive value of lactate concentration is debatable. In addition, most studies on the correlation between lactate and DNS development were conducted at a single center with a small sample size. Therefore, further study is needed to synthesize and interpret the results of serum lactate concentration for DNS prediction in acute CO poisoning. This systematic review and meta-analysis aimed to investigate the relationship between the initial serum lactate levels and DNS development in patients with acute CO poisoning.

## 2. Materials and Methods

### 2.1. Protocol and Registration

This systematic review and meta-analysis was conducted according to the Preferred Reporting Items for Systematic Reviews and Meta-Analysis (PRISMA) and Meta-analysis of Observational Studies in Epidemiology (MOOSE) guidelines [33,34]. The review protocol is registered at https://www.crd.york.ac.uk/prospero/display_record.php?ID=CRD42021243025 (accessed on 15 April 2021). Institutional review board approval and informed consent was not required for this meta-analysis.

### 2.2. Eligibility Criteria

A questionnaire framework based on the population, intervention, comparison, and outcome (PICO) was applied in this study. A literature search on critical assessments was performed to draft a summary of the eligible studies, and their outcomes were then evaluated through a meta-analysis. The PICO questions were as follows: population (P) = adult patients with acute CO poisoning; intervention (I) = serum lactate level in the early phase upon admission in the emergency department; comparator (C) = none; outcome (O) = occurrence of DNS.

### 2.3. Information Sources and Literature Search Strategy

We performed an extensive database search to identify all relevant studies that examined the role of serum lactate levels in predicting DNS in patients with acute CO poisoning. The search encompassed the EMBASE (1974 to 5 January 2022) and MEDLINE (1946 to 5 January 2022) databases via the Ovid interface. Databases (KoreaMED, KMBASE, KISS, NDSL, KISTi, and RISS) related to Korea and Google Scholar were also selected for the analysis. Additionally, we manually cross-referenced the eligible studies to identify other relevant studies. Two experienced reviewers (HL and JO) conducted the latest update to our search on 5 January 2022. The following search terms were used: “lactate” and “carbon monoxide” (Appendix A). No language restrictions or methodological filters were used, and prospective or retrospective observational studies were selected from the list.

### 2.4. Study Selection

Two experienced reviewers (HL and JO) independently screened titles and abstracts to filter irrelevant studies. The following criteria were used for exclusion of studies: irrelevant outcomes, irrelevant intervention, irrelevant populations, irrelevant article type (reviews, case reports, editorials, letters, comments, conference abstracts, animal studies, and meta-analyses) and duplicated data. In case of disagreement between the two reviewers, a third reviewer (HK) was allowed to intervene, and the differences in opinion were discussed until a consensus was reached. After excluding extraneous abstracts, the full texts of the selected studies were re-screened and reviewed thoroughly for eligibility using the predetermined selection criteria. Studies with insufficient data despite contacting the authors were also excluded. Finally, prospective or retrospective observational studies on patients who presented to EDs for CO poisoning with elevated serum lactate levels, where the serum lactate was collected after admission and, also, examined with developed DNS, were included in this systematic review and meta-analysis.

### 2.5. Data Collection Process and Data Items

The basic characteristics and main results of the selected studies were extracted by the two reviewers (HL and JO). Any disagreements between the reviewers were resolved by consensus. The study characteristics and extracted covariates were summarized using standard descriptive statistics. Dichotomous variables were reported as frequencies (%), whereas continuous variables were reported as means (standard deviation [SD]). The following variables were extracted: study number, author, year of publication, country, inclusion period, study design, inclusion criteria of each study in this meta-analysis, the timing of lactate measurement, sample size, age, sex, the proportion of HBOT management, the definition of DNS and observation period of DNS occurrence from hospital discharge. The mean (±SD) serum lactate level and measurement units were also recorded, and the estimated mean (±SD) values were calculated from the median values with interquartile ranges [35]. The unit with the highest frequency in the included studies was selected, and the other units were converted to selected units.

### 2.6. Risk of Bias in Individual Studies

Two reviewers (HL and JO) independently evaluated the methodological integrity of the included studies, where the authors and journals were blinded to the reviewers. Six bias domains (study participation, study attrition, prognostic factor measurement, outcome measurement, study confounding, statistical analysis and reporting) were assessed using the QUIPS tool in systematic reviews [36]. To determine the overall risk, studies with a low risk of the six abovementioned bias domains were rated as high-quality studies [36].

### 2.7. Statistical Analysis

In the meta-analysis, we estimated the association between serum lactate levels in the early phase in adult patients with acute CO poisoning and the development of DNS. The strength of the association between elevated serum lactate levels and DNS was estimated using standardized mean differences (SMD). A random-effects model was used to interpret the individual data of the included studies, considering the diversity of countries, inclusion periods, inclusion criteria of each study and timing of lactate measurement.

To measure heterogeneity, I^2^ statistics were used to estimate the proportion of inter-study inconsistency due to true differences between studies (rather than differences due to random error or chance), with values of 0–40%, 30–60%, 50–90% and 75–100% denoted as “might not be important”, “may represent moderate heterogeneity”, “may represent substantial heterogeneity” and “considerable heterogeneity”, respectively [37,38].

The reference management software Endnote X9 (Clarivate Analytics LLC, Philadelphia, PA, United States) was used to organize all studies identified in the literature search. We also used RevMan version 5.4.1 (Cochrane Collaboration, Nordic Cochrane Centre, Copenhagen, Denmark) and R version 4.0.4 (R Foundation for Statistical Computing, Vienna, Austria) statistical software to perform the statistical analysis, where a *p*-value of <0.05 was considered as statistically significant.

### 2.8. Additional Analyses

We performed planned subgroup analysis for the following confounders to identify heterogeneity: proportion of patients managed with HBOT (all patients (100%) vs. partial patients (<100%)), the country where the study was performed (Korea vs. others), quality of the study according to the QUIPS tool (high-quality study vs. low-quality study), sample size according to the median value across the included studies (large sample size vs. small sample size). Sensitivity analysis was conducted by sequentially omitting studies to interpret the potential causes of heterogeneity between the studies. Meta regression analysis was performed to identify heterogeneity and analyze the effect of study characteristics on the results. The asymmetry of the contour-enhanced funnel plot was investigated to identify publication bias. The results were considered statistically significant at *p* < 0.05.

### 2.9. Level of Evidence

The level of evidence was graded as high, moderate, low, and very low using the Grading of Recommendations, Assessment, Development, and Evaluation (GRADE) framework [39]. It was conducted using GRADEProfier (version 3.6.1, The GRADE Working Group), and a summary of findings was presented by an evidence profile.

## 3. Results

### 3.1. Study Selection

A flow diagram of the literature search for this systematic review is presented in Figure 1. A total of 606 records were identified using the database, along with an additional manual search. A total of 234 duplicates were removed, and an additional 290 irrelevant records were excluded based on titles and abstracts. After reviewing the full texts of the 82 remaining records, we excluded 74 records, including irrelevant population (*n* = 13), irrelevant outcomes (*n* = 31), irrelevant intervention (*n* = 26), duplicated data from the same studies (*n* = 1) and irrelevant articles (*n* = 3). Finally, eight observational studies that enrolled 1350 patients were included in this meta-analysis [17,18,19,20,21,22,23,24].

### 3.2. Study Characteristics

The main attributes of the included studies are presented in Table 1. Additionally, the baseline characteristics of the enrolled patients are provided in Appendix A, and the DNS definition of each study is summarized in Appendix A. Eight observational studies were published between 2011 and 2021. Six studies were conducted in Korea, whereas the remaining studies were conducted in Turkey and Italy. All studies were single center studies, and four studies were prospectively designed. The inclusion criterion for all studies was the presence of COHb, where the cut-off was 3% or 5%. In seven studies, lactate levels were measured at the time of admission of patients to ED, and in one study, the level was measured within 6 h of admission to ED. The observation period for the occurrence of DNS after hospital discharge spanned from 6 weeks to a year, and 22.6% of patients with CO poisoning developed DNS. The proportion of patients managed with HBOT was 83.3%, varying from 22.1% to 100%.

### 3.3. Risk of Bias within Studies

The risk of bias for the eight included studies was assessed using the QUIPS tool (Appendix A). The risk of bias was assessed as high in five studies. The major cause of high risk originated from outcome measurement caused by a lack of objectivity in the DNS diagnostic criteria in four studies. One study was assessed as having a high risk of bias in terms of study participation, study attrition and statistical analysis and presentation.

### 3.4. Results of Meta-Analyses

#### 3.4.1. Serum Lactate Level and Occurrence of Delayed Neuropsychiatric Sequelae

In this meta-analysis, the early phase serum lactate level was significantly higher in the DNS group than in the non-DNS group, with moderate heterogeneity (8 studies; SMD, 0.31; 95% CI, 0.11–0.50; I^2^ = 44%; *p* = 0.002; Figure 2). Four studies showed no differences between the DNS and non-DNS groups, and four studies showed significantly higher lactate levels in the DNS group.

#### 3.4.2. Additional Analysis for Identifying and Measuring Heterogeneity

The results of the predefined subgroup analyses are summarized in Appendix A. The differences between the subgroups were not significant for the three characteristics except HBOT (I^2^ = 0%). However, the subgroups by the proportion of HBOT therapy did not show decreased heterogeneity. A summary of the sensitivity analysis is presented in Figure 3 and Appendix A. The heterogeneity decreased to “might not be important” after omitting Han 2021 (7 studies; SMD, 0.38; 95% CI, 0.23–0.53; I^2^ = 8%; *p* < 0.001). No significant reduction in heterogeneity was seen after omitting seven other studies.

There was no definite asymmetry in the forest plots. No significant asymmetry was confirmed in the results of the contour-enhanced funnel plot, which was used to evaluate reporting bias, such as publication bias (Appendix A). Moreover, there was no significant influence of the observation period of DNS on the results of meta-regression analysis (*p* = 0.12; Appendix A).

#### 3.4.3. Level of Evidence

The results of this study were assigned a low level of evidence according to the evidence profile using the GRADE framework (Appendix A). Analysis of observational studies was the main reason for the low level of evidence. The importance of the result was judged as critical because DNS is a severe sequela in patients, where the prediction is crucial for prevention.

## 4. Discussion

The prediction of DNS development in patients with CO poisoning at an early phase is an important factor for treatment; however, tools for accurate prediction are insufficient. Several studies have investigated biomarkers as predictors of DNS. However, most of these studies were conducted at a single center, where the sample size was relatively small, and the results were inconsistent.

In this study, we investigated the association between early phase serum lactate levels and the occurrence of DNS in patients with acute CO poisoning using meta-analysis. We found that patients who developed DNS had considerably higher serum lactate levels in the early phase of acute CO poisoning than those who did not develop DNS. To the best of our knowledge, no study has been identified in which that investigated the relationship between biomarkers and DNS occurrence was investigated using meta-analysis.

Normal blood lactate concentration is approximately 1 mEq/L, and even a small increase in lactate concentration to 1.5 mEq/L or higher in critically ill patients is associated with high mortality [40,41]. The serum lactate concentration has been widely used for many years as an indicator of changes in tissue perfusion in critically ill patients [30,31,42]. In addition to the increase in anaerobic metabolism, hyperlactatemia is caused by increased glycolysis, catecholamine-stimulated Na^+^–K^+^ pump activity, altered pyruvate dehydrogenase activity, and decreased lactate clearance due to hepatic hypoperfusion [43].

Serum lactate concentration could be enhanced in CO poisoning due to anaerobic glycolysis leading to systemic hypoxia; a relationship between serum lactate concentration and the severity of CO poisoning has been reported [32,44,45,46]. Cervellin et al. reported a significant correlation between the initial blood COHb level and lactate level in patients with CO poisoning (r = 0.54; *p* < 0.001), where lactate level was useful as a crucial indicator in predicting the need for hospitalization of patients with CO poisoning [32]. Another retrospective study reported that lactate level could be used as an important factor for the prediction of severe complications and the need for intensive care unit treatment, along with old age, white blood cell count and level of consciousness at hospitalization [44]. However, Benaissa et al. reported that the serum lactate level alone was insufficient to measure the degree of CO poisoning in a prospective study of 146 patients with CO poisoning [45]. Another study reported that the initial lactate level in CO poisoning could be considered an adjunctive parameter of severity along with the clinical criteria and COHb [46].

Studies comparing the initial blood lactate concentrations of the DNS and non-DNS groups have also been reported. In a retrospective study conducted by Pepe et al., there was no significant difference in the initial blood lactate concentration between DNS and non-DNS groups (DNS vs. non-DNS, 1.77 vs. 1.76; *p* > 0.05) [17]. Zhang et al. reported a significant difference in lactate concentration elevation between DNS and non-DNS groups in univariate analysis (DNS vs. non-DNS, 38% vs. 13%; *p* = 0.008), whereas no significant difference was seen in multivariate analysis (*p* = 0.12) [47]. Moreover, it has also been suggested that the initial lactate concentration did not help predict DNS in multivariate analyses conducted in other studies [22,24].

Elevated lactate levels in the early phase of CO poisoning can lead to systemic cellular ischemia and inflammatory response due to the direct effect of CO. Although it is difficult to assume that enhanced lactate levels specifically cause damage to the central nervous system, it is highly likely that the damage increases proportionally with hypoxia and abnormal inflammation of the central nervous system, considering the characteristics of CO poisoning. In addition, there is a correlation between acute-phase brain damage and high lactate concentration, considering the marked association between high blood lactate concentration and changes in consciousness in acute CO poisoning [44]. Acute brain nerve damage is anticipated to induce DNS through mechanisms such as lipid oxidation and excess dopamine during the recovery phase.

In this study, the association between serum lactate levels in the early phase of CO poisoning and the occurrence of DNS was analyzed in eight included studies. There was a significant difference in the early phase serum lactate concentration between the DNS and non-DNS groups with a moderate degree of heterogeneity. In the sensitivity analysis, heterogeneity decreased to I^2^ = 8% from I^2^ = 44%, omitting the results of the study conducted by Han et al. [19]. They reported that a shift of a large proportion of the included patients from other hospitals that received oxygen therapy led to a difference in the biomarker level between the time of poisoning and time of blood sample collection [19]. The heterogeneity with other studies could be attributed to the fact that the lactate level is highly affected by oxygen treatment, which may have influenced their results.

This study has several limitations. First, there was heterogeneity among the studies included in the meta-analysis and patient characteristics. This study was conducted using SMD, a random-effects model considering variation among studies and additional analysis was performed to decrease heterogeneity. Second, five out of the eight included studies were assessed as low quality. Third, there are no universal guidelines or diagnostic criteria for DNS, and the clinical manifestations of DNS vary from mild to severe. Therefore, different diagnostic criteria for each study may have influenced the results and heterogeneity. Fourth, six of the eight included studies were conducted in Korea. The results of this study cannot be extrapolated to other racial groups and countries worldwide. Fifth, the time from CO exposure to blood sampling and follow-up or changes in lactate levels was not included in the analysis. Sixth, we did not consider confounders that might affect the lactate levels, such as underlying diseases, route of CO exposure, intentionality, use of drugs or alcohol and smoking history.

## 5. Conclusions

The early phase serum lactate concentration was significantly higher in the DNS group than in the non-DNS group in adult patients with acute CO poisoning. A lactate concentration test for patients with CO poisoning could help predict DNS. A well-designed, large-scale, prospective study is required to support the results of this study due to the presence of less evidence in the literature.

## Figures and Tables

**Figure 1 jpm-12-00651-f001:**
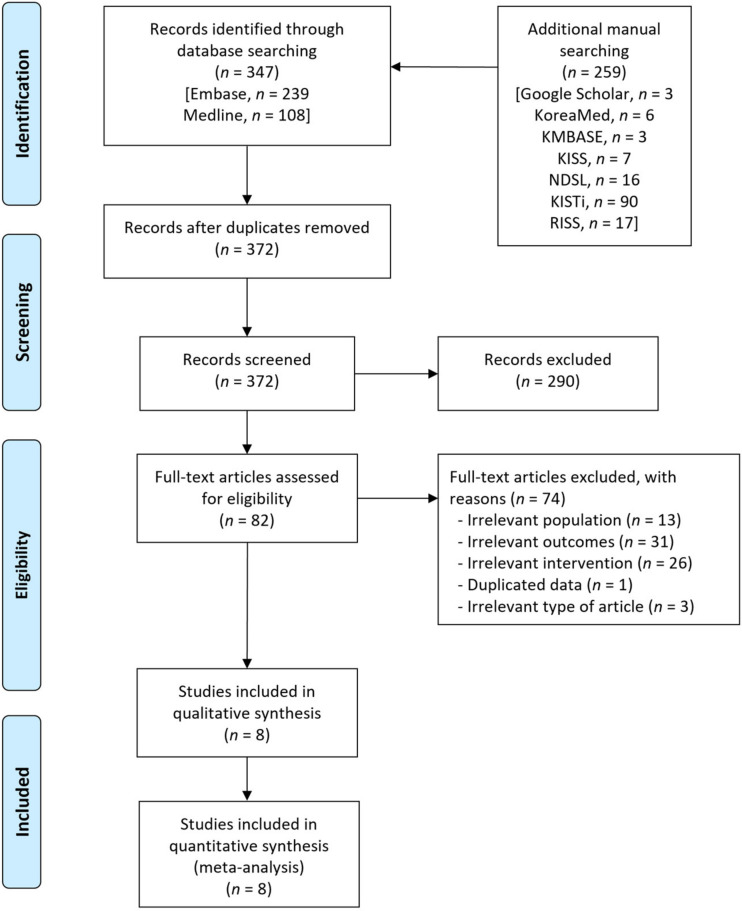
Flow diagram for the identification of relevant studies.

**Figure 2 jpm-12-00651-f002:**
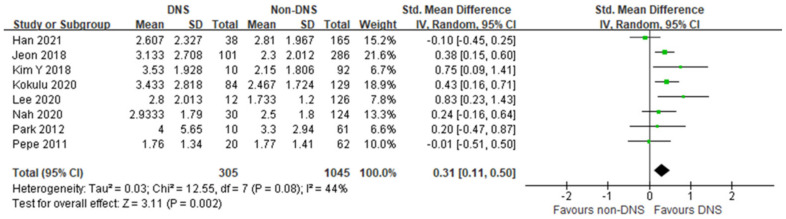
Forest plot of the association between early phase serum lactate level and occurrence of delayed neuropsychiatric sequelae in adult patients with acute carbon monoxide poisoning [17,18,19,20,21,22,23,24].

**Figure 3 jpm-12-00651-f003:**
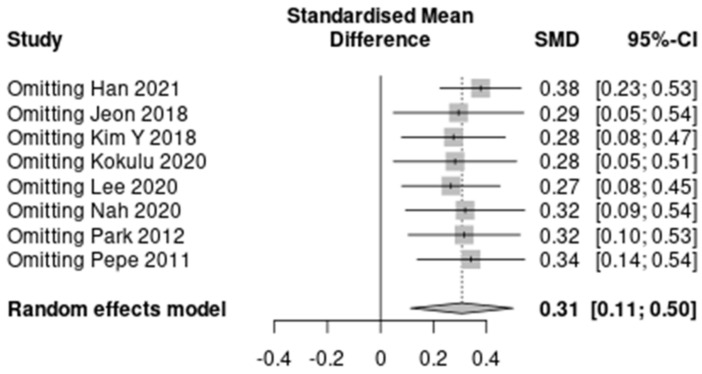
A summary of the sensitivity analysis [17,18,19,20,21,22,23,24].

**Table 1 jpm-12-00651-t001:** Study characteristics included in the meta-analysis.

Study	Region	Period	Design	Inclusion Criteria	Timing of Laboratory Examination	Number of Pts. DNS/Total (%)	Observation Period of DNS Occurrence from Hospital Discharge
Han 2021 [19]	Korea	Jul 2017–Feb 2020	sPOS	COHb ≥ 5% (Smokers: ≥10%)	At ED arrival	38/203 (18.7)	3 months
Jeon 2018 [20]	Korea	Apr 2011–Dec 2015	sPOS	Acute COP	At ED arrival	101/387 (26.1)	6 weeks
Kim Y 2018 [21]	Korea	Jan 2015–May 2016	sROS	COHb ≥ 5% (Smokers: ≥10%)	At ED arrival	10/102 (9.8)	2 months
Kokulu 2020 [22]	Turkey	Aug 2018–Jul 2019	sPOS	COHb ≥ 5% (Smokers: ≥10%)	At ED arrival	54/183 (29.5)	6 weeks
Lee 2021 [24]	Korea	Jan 2018–Jul 2018	sROS	COHb ≥ 3% (Smokers: ≥10%)	At ED arrival	12/138 (8.7)	6 weeks
Nah 2020 [18]	Korea	Aug 2016–Jul 2019	sPOS	COHb ≥ 5% (Smokers: ≥10%)	At ED arrival	30/154 (19.5)	3 months
Park 2012 [23]	Korea	Mar 2011–Sep 2011	sROS	COHb ≥ 3% (Smokers: ≥10%)	At ED arrival	10/71 (14.1)	6 months
Pepe 2011 [17]	Italy	1992–2007	sROS	COHb > 5% (Smokers: >10%)	Within 6 h from ED arrival	34/141 (24.1)	1 year

Abbreviations: COHb, carboxy hemoglobin; DNS, delayed neuropsychiatric sequelae; ED, emergency department; Pts., patients; sPOS, single-center prospective observational study; sROS, single-center retrospective observational study.

## Data Availability

Not applicable.

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
