# Peer review of "Association between Early Phase Serum Lactate Levels and Occurrence of Delayed Neuropsychiatric Sequelae in Adult Patients with Acute Carbon Monoxide Poisoning: A Systematic Review and Meta-Analysis"

_jpm, 2022, doi:10.3390/jpm12040651_

Round 1
Reviewer 1 Report
The authors performed a meta-analysis and found that early phase serum lactate concentration was significantly higher in the DNS group than in the non-DNS group in adult patients with acute CO poisoning. The authors concluded that patients' lactate concentration tests for CO poisoning could help predict DNS. The paper is well written, and the rationale is very clear. The selection of potential studies is clear and well-reasoned. Potential heterogeneity and sensitivity analyses are carefully, and the study's limitations were well acknowledged. My primary concern is the time of blood withdrawal to perform lactate measurements. Authors should check when the blood was withdrawn (immediately after hospitalization or after the diagnosis of DNS). Also, it would be great to discuss the findings of longitudinal studies if published even in any other population. The occurrence of DNS varies among different studies and thus could be a confounder. Did the author include the time of DNS occurrence in the analysis? Smoking causes both immediate and long-standing effects on serum lactate levels. Did the author check the influence of smoking on this meta-analysis's findings? It is unclear whether the DNS is the categorical outcome or the continuous outcome. What tests were performed in the studies to assess the DNS?
Author Response
Dear Reviewer:
Thank you for your comments. We agree with your opinion about our study and manuscript.
We have made several changes in the manuscript in accordance with your suggestions.
We acknowledge that your insightful comments have helped us correct the errors and enhance the quality of the manuscript.
Our point-by-point responses to your comments are given below.
Best regards.

Reviewer 2 Report
General comment:
This is an informative and important study. I had only few comments to be addressed.
Abstract:
- The authors omit the introduction in their abstract and jump into method section. I would recommend the authors to provide a brief statement about the rationale of this study.
- The lactate levels was significantly higher in DNS group than in non-DNS group did not guarantee “A lactate concentration test for patients with CO poisoning could help predict DNS”. The statistical result of a meta-analysis could only guarantee “The DNS group was associated with significantly higher lactate concentration than that in non-DNS group”. Please re-write this statement accordingly.
Introduction:
- The authors said “COHb is used as a standard indicator to confirm the severity of CO poisoning; however, it does not help predict DNS [17,26,27]” and then said “Since the major effect of CO is systemic hypoxia, accompanied by an increase in lactate concentration, elevated serum lactate concentration can be used as an indicator of systemic hypoxia [28–30]. Accordingly, several previous studies have been conducted to compare the initial lactate concentration in DNS and non-DNS development groups, where inconsistent results have been reported, with remarkable differences [17–24]”. I would recommend the authors to insert one-two statements about why the lactate concentration could serve to predict DNS/non-DNS development. Is there any different predictive value between COHb and lactate?
- The authors cited reference [17–24] in the introduction section to say that there is inconsistent result. I would recommend the authors to briefly mentioned the results of these studies.
Method:
- The authors listed several extra databases (KoreaMED, KMBASE, KISS, NDSL, KISTi, and RISS) in the method section. It would be great. However, I would recommend the authors to also list them in the abstract section.
- The authors said “with values of 0–40%, 30–60%, 50–90%, 75–100%” to be the cut-off point of heterogeneity. I have a question. How do the authors define if the I2 value is 55%? moderate heterogeneity? substantial heterogeneity?
- The authors arranged subgroup analysis (i.e. additional analysis) in their study. This would be great. However, it would be helpful if the authors provide the rationale of the cut-off point of their subgroup. For example, proportion of patients managed with HBOT (>90% vs. <90%) and sample size (large sample size vs. small sample size). In addition, how did they define high-quality/low-quality of the study and large/small sample size. In addition, please provide the rationale of the definition of high-quality/low-quality of the study and large/small sample size.
Result:
- In the figure 1, the authors also searched on Google scholar, which was not listed in method. Please clarify this.
Author Response

(The authors gave the same response as above.)

Round 2
Reviewer 2 Report
Many thanks to addressing all my comments. I had no further comments for this manuscript. The manuscript is acceptable for the current version